# PET/CT and Conventional Imaging for the Assessment of Neuroendocrine Prostate Cancer: A Systematic Review

**DOI:** 10.3390/cancers15174404

**Published:** 2023-09-03

**Authors:** Francesco Dondi, Alessandro Antonelli, Nazareno Suardi, Andrea Emanuele Guerini, Domenico Albano, Silvia Lucchini, Luca Camoni, Giorgio Treglia, Francesco Bertagna

**Affiliations:** 1Nuclear Medicine Department, University of Brescia and ASST Spedali Civili di Brescia, 25123 Brescia, Italy; 2Department of Urology, AOUI Verona, University of Verona, 37134 Verona, Italy; 3Department of Urology, Spedali Civili di Brescia, 25123 Brescia, Italy; 4Department of Radiation Oncology, University of Brescia and ASST Spedali Civili di Brescia, 25123 Brescia, Italy; 5Nuclear Medicine, Imaging Institute of Southern Switzerland, Ente Ospedaliero Cantonale, 6500 Bellinzona, Switzerland; 6Department of Nuclear Medicine and Molecular Imaging, Lausanne University Hospital, University of Lausanne, 1011 Lausanne, Switzerland; 7Faculty of Biomedical Sciences, Università della Svizzera Italiana, 6900 Lugano, Switzerland

**Keywords:** neuroendocrine prostate cancer, NEPC, positron emission tomography, PET/CT, conventional imaging

## Abstract

**Simple Summary:**

Neuroendocrine prostate cancer is a rare neoplasm with aggressive behavior and poor prognosis. Its diagnostic approach is challenging since no specific features are present when using conventional imaging or positron emission tomography (PET). The aim of this systematic review was, therefore, to evaluate the role of these imaging modalities for the assessment of neuroendocrine prostate cancer. At present, it is still uncertain which tracer performs best, and although [^18^F]FDG has been evaluated and seems to offer some advantages in availability and clinical staging, other tracers may be more useful to understand tumor biology or identify targets for subsequent radioligand therapy. Further research is therefore desirable. In contrast, data are still limited to draw a final conclusion on the role and the specific characteristics of CI in this rare form of neoplasm, and therefore, more studies are needed in this setting.

**Abstract:**

Background: Neuroendocrine prostate cancer (NEPC) is a rare neoplasm, and the role of both conventional imaging (CI) and positron emission tomography/computed tomography (PET/CT) for its assessment has not been clearly evaluated and demonstrated. The aim of this systematic review was to analyze the diagnostic performances of these imaging modalities in this setting. Methods: A wide literature search of the PubMed/MEDLINE, Scopus, and Web of Science databases was made to find relevant published articles about the role of CI and PET/CT for the evaluation of NEPC. Results: 13 studies were included in the systematic review. PET/CT imaging with different radiopharmaceuticals has been evaluated in many studies (10) compared to CI (3 studies), which has only a limited role in NEPC. Focusing on PET/CT, a study used [^18^F]FDG, labeled somatostatin analogs were used in 5 cases, a study used [^68^Ga]Ga-FAPI-04, [^68^Ga]Ga-PSMA-11 was evaluated in a single case, and two works used different tracers. Conclusion: Published data on the role of PET/CT for the assessment of NEPC are limited. At present, it is still uncertain which tracer performs best, and although [^18^F]FDG has been evaluated and seems to offer some advantages in availability and clinical staging, other tracers may be more useful to understand tumor biology or identify targets for subsequent radioligand therapy. Further research is therefore desirable. In contrast, data are still limited to draw a final conclusion on the role and the specific characteristics of CI in this rare form of neoplasm, and therefore, more studies are needed in this setting.

## 1. Introduction

Neuroendocrine prostate cancer (NEPC) is a rare type of prostate cancer (PCa) that can arise de novo from normal prostatic neuroendocrine cells that undergo oncogenic mutation (0.5–2% of the cases) or can derive after neuroendocrine transdifferentiation of adenocarcinoma in the case of castration-resistant prostate cancer (CRPC) (17–20%) [1,2,3]. While the first form is particularly rare, this latter type is characterized by the presence of particular molecular changes that can make it resistant to traditional castration therapy, including androgen receptor-targeted drugs [4]. In the presence of metastatic CRPC, treatment-emergent NEPC can be observed as the development of resistance to androgen deprivation therapy (ADT) and after progression, while taking potent androgen receptor signaling inhibitors, such as abiraterone and enzalutamide. In this setting, previous analyses indicated that the treatment-emergent form of NEPC detected using immunohistochemical staining is present in approximately 20% to 30% of metastatic castration-resistant tumors [2]. Even if the exact pathogenesis of neuroendocrine differentiation in the human prostate is still poorly understood, it is presumed that no single pathway is involved in this phenomenon, and many different mechanisms of ADT resistance have been proposed, such as the development of genetic changes that restore the presence of androgen receptor (AR) signaling even when androgen levels are low (including genomic amplification), mutation that convert antiandrogens into agonists and splice variants of AR that can activate them independently from the presence of a specific ligand. Moreover, upregulation of glucocorticoid receptor levels and activity can bypass the blockade of AR and lineage switching in letting tumor cells acquire phenotypic characteristics typical of a cell lineage whose survival does not depend on the drug target [1].

Speaking of epidemiology, it is thought that the incidence of NEPC could be higher than what is really observed, given the recent introduction of new androgen signaling inhibitors, the limited number of biopsies performed, and the frequent misclassification of NEPC as high-grade prostate adenocarcinoma [2]. Furthermore, a mixed form of adenocarcinoma and NEPC can be present [5].

From a clinical point of view, NEPC is far more aggressive than adenocarcinoma, and pure NEPCs are rapidly more symptomatic, locally advanced, or metastatic at the time of diagnosis, with the frequent presence of visceral metastases [6,7,8,9,10]. In this scenario, the prognosis of patients affected by NEPC is extremely poor, with a median estimated survival of around 10 months [11]. Most patients with this neoplasm die within 1 to 2 years from the diagnosis. Interestingly, in the setting of localized disease, an increased proportion of neuroendocrine differentiation confers an adverse prognosis independent from the Gleason grade or tumor stage [2]. In this setting, the main clinical presentations of NEPC include androgen deprivation resistance, low levels of prostate-specific antigen (PSA), the disproportion between PSA kinetic and tumor burden progression, and the eventual increase in neuroendocrine tumor markers [3,12]. These markers can be useful for the diagnosis and the evaluation of the transdifferentiation of the disease; chromogranin A (CgA) and neuron-specific enolase (NSE) reflect its neuroendocrine nature and could also be used as prognostic factors [13,14]. Several published studies have reported a correlation between CgA and NSE serum levels, androgen independence, progression of the disease, and prognosis. [9]. Although still not well defined, the reasons that might cause the poor prognosis of NEPC may include active neuroendocrine cell production of growth factor and the lack of AR in such cells, which would also account for unresponsiveness to hormonal treatment, as mentioned. In patients with castration-resistant tumors, elevated serum levels of CgA are significant predictors of poor prognosis independently from serum PSA levels. Moreover, pretreatment measurement of CgA and NSE levels can predict prognosis after hormone therapy [13].

There are many clues that can suggest the presence of NEPC at presentation, such as the limited temporal response to primary ADT (<6 months), high PSA nadir on ADT, the presence of visceral metastases (including lungs, liver, and central nervous system), the predominant presence of lytic bone metastases, low absolute PSA levels compared to the burden of the disease and the elevation of the aforementioned neuroendocrine serum markers [2]. Even if these clinical features may be suggestive of the presence of NEPC, the final diagnosis currently remains based on histopathology. Therefore, biopsy should be strongly considered whenever clinical features and/or serum markers are suggestive of neuroendocrine differentiation. In this setting, standard pathologic evaluation should quantify the level of expression of the aforementioned markers and determine if small cell features are present. As previously underlined, a diagnosis of neuroendocrine differentiation has both prognostic and therapeutic implications from a pathological point of view [2].

Speaking of diagnosis, conventional imaging (CI) such as magnetic resonance (MR), computed tomography (CT), and ultrasound (US) are useful for staging and restaging of classic PCa, even if their role in NEPC could be limited and not specific [15,16]. In particular, CI cannot directly differentiate between NEPC and other prostatic neoplasms, but this imaging modality can reflect the presence of aggressive forms. In this setting, the presence of necrosis and hemorrhage, local invasion, nodal and visceral metastases, and their rapid progression may suggest the presence of NEPC, which often requires histopathologic confirmation, as stated before [17].

Focusing on therapy, the treatment of NEPC relies mainly on the use of systemic therapy, such as chemotherapy with carboplatin and cabazitaxel, which are the main therapeutic agents used in the first-line regimen [5,18]. Since this neoplasm is a highly proliferative subset of PCa, it frequently responds rapidly to cytotoxic chemotherapy with minimal or absent response to ADT, as previously underlined [2]. Furthermore, if intensive neuroendocrine differentiation is identified, it should be considered milder ADT, such as intermittent androgen deprivation or antiandrogen monotherapy, to slow this differentiation and/or neuroendocrine-targeted therapy. Potential therapies directed toward neuroendocrine hormones and/or their antagonists, such as somatostatin, bombesin, and serotonin, have been receiving attention in the past, however, without clear evidence of their therapeutic role [13].

Generally speaking, it is known that nuclear medicine has a prominent role in the assessment of PCa, in particular with the use of positron emission tomography/computed tomography (PET/CT) and bone scintigraphy [19]. In the first case, prostate-specific membrane antigen (PSMA) labeled with both [^18^F] or [^68^Ga] is a radiopharmaceutical that has proved its high diagnostic accuracy for the evaluation of PCa and enables the selection of patients that could be treated with [^177^Lu]Lu-PSMA-617. Furthermore, this imaging modality can also be used for the evaluation of different medical conditions not related to the prostate, both malignant and benign [3,20,21,22]. As mentioned, NEPC can evolve from classical PCa with a neuroendocrine transdifferentiation process, and therefore, different tracers have been used to study such neoplasm. In this scenario, [^18^F]fluorodeoxyglucose ([^18^F]FDG), a radiolabeled glucose analog and the most used radiopharmaceutical worldwide, is a tracer able to evaluate a wide range of tumors, and in PCa, it can reflect the possible resistance to castration therapy that can lead to the presence of NEPC, even if clear evidence has not been demonstrated in this setting [3,23,24]. Interestingly, the use of PET/CT with radiolabeled somatostatin analogs in PCa and in particular for NEPC, has been investigated in order to eventually propose peptide receptor radionuclide therapy (PRRT) for these patients [25,26].

The aim of this systematic review is, therefore, to evaluate the role of both CI and PET/CT, performed with different tracers, for the assessment of NEPC.

## 2. Materials and Methods

### 2.1. Search Strategy

A wide literature search of the PubMed/MEDLINE, Scopus, and Web of Science databases was performed in order to find significant published articles concerning the role of PET/CT and CI for the assessment of NEPC. The algorithm used for the research was the following: “neuroendocrine” AND “prostate” AND (“PET” OR “positron emission tomography” OR “MR” OR “magnetic resonance” OR “CT” OR “computed tomography” OR “imaging” OR “diagnosis” OR “US” OR “ultrasound” OR “staging”).

No beginning date limit was applied to the search, and it was updated until 1 March 2023. Preclinical studies, conference proceedings, case reports with only one patient, reviews, or editorials were excluded. To expand our search, the references of the retrieved articles were also screened for additional papers.

### 2.2. Study Selection

Two researchers (F.D. and D.A.) independently reviewed the titles and abstracts of the retrieved articles. The same two researchers then independently reviewed the full-text version of the remaining articles to determine their eligibility for inclusion.

### 2.3. Quality Assessment

The quality assessment of these studies, including the risk of bias and applicability concerns, was carried out using the Quality Assessment of Diagnostic Accuracy Studies version 2 (QUADAS-2) evaluation [27].

### 2.4. Data Extraction

For each study included in the review, data concerning authors’ names, year of publication, country of origin, design of the study, radiopharmaceuticals used if applicable, number of patients, type of scan and tomograph used, and the setting were collected. The main findings of the articles are reported in the Results section.

## 3. Results

### 3.1. Literature Search

A total of 3427 articles were extrapolated with the computer literature search, and by reviewing the titles and abstracts, 3414 of them were excluded because the reported data were not within the field of interest of this review. Thirteen articles were therefore selected and retrieved in full-text version [28,29,30,31,32,33,34,35,36,37,38,39,40]; no additional studies were found screening the references of these articles (Figure 1).

In general, the quality assessment using QUADAS-2 evaluation underlined the presence of a high risk of bias and applicability concerns in some of these studies for patient selection, and this is mainly related to the fact that these studies were characterized by heterogeneous cohorts (Figure 2).

Among the total number of studies included in the systematic review, 10 were of a retrospective nature [28,29,33,34,35,36,37,38,39,40], whereas 2 had a prospective design [28,29], and in 1 case, it was not specified its nature [32]. Ten studies focused on PET/CT imaging [28,37], while three studies were performed using CI [38,39,40].

Speaking about radiopharmaceuticals used for PET/CT imaging, 1 of the study was performed with [^18^F]FDG [29], in 3 cases [^68^Ga]Ga-DOTANOC was used [28,30,31], in 2 cases [^68^Ga]Ga-DOTATATE was used [32,36], [^68^Ga]Ga-FAPI-04 was used in 1 case [34], 1 study was performed with [^68^Ga]Ga-PSMA-11 [35] and lastly two works used different tracers: 1 with [^68^Ga]Ga-PSMA, [^68^Ga]Ga-DOTATATE and [^18^F]FDG [33] and 1 with [^18^F]-PSMA-1007, [^18^F]-AIF-NOTA-octreotide and [^18^F]FDG [37].

The main characteristics of the studies and their results are briefly presented in Table 1.

### 3.2. PET/CT Studies

As previously underlined, different studies evaluated the value of PET/CT imaging for the assessment of NEPC [28,29,30,31,32,33,34,35,36,37], demonstrating some insights into its possible usefulness.

#### 3.2.1. [^18^F]FDG

Speaking about [^18^F]FDG PET/CT, its role in NEPC was first evaluated by Spratt et al. [29], revealing the presence of tracer-avid lesions in 15/23 patients. Interestingly, PET was able to demonstrate 5.4% and 6.8% of the lesions that were not detected with CT or bone scan, respectively. For soft tissue lesions, 95.1% of them were demonstrated using hybrid imaging, with PET that was also able to underline a lesion not detected using CT. Notably, an average standardized uptake value (SUVmax) of 4.52 for bone lesions and 6.65 for soft tissue lesions were reported. In a prognostic setting, patients with more bone and soft tissue avid lesions at PET/CT (*p* values 0.06 and 0.01, respectively) or higher average SUVmax at the bone and soft tissue lesions (*p* value 0.04 and <0.01, respectively) had lower survival. Interestingly, no correlation between PET results and serum marker levels was reported.

#### 3.2.2. Radiolabeled Somatostatin Analogs

The first study in this setting was performed by Fanti et al. [28], who evaluated different unusual neuroendocrine tumors comprising 3 NEPC with [^68^Ga]Ga-DOTANOC. PET/CT was true positive in 1/3 cases and true negative in 2/3 cases.

[^68^Ga]Ga-DOTANOC PET/CT was more recently used by Savelli et al. [30] to evaluate 2 cases of NEPC, reporting that it had the capability to visualize bone metastatic lesions, in particular for lytic ones that had higher uptake, but also lymphangitic neoplastic spread to the lungs. In subsequent work, the same authors evaluated six patients with CRPC again with [^68^Ga]Ga-DOTANOC PET/CT, revealing in 2 of them the presence of bone or lung metastases, however, with scant uptake (SUVmean 1.57); interestingly, a case of false negative scan in comparison with [^18^F]-choline was reported [31]. Similarly, somatostatin receptor expression in CRPC was evaluated by Gofrit et al. [32] with [^68^Ga]Ga-DOTATATE, confirming its ability to detect both blastic and lytic bone metastases with a moderately high tracer uptake on most of the blastic ones (mean SUVmax for blastic lesions 5.3, 7.2 for lytic or nodal lesions). Moreover, the authors reported that patients with multiple bone lesions had a significantly higher SUVmax compared with patients with few metastases (*p* value 0.05), but, despite that, only a low correlation was reported between the degree of uptake and PSA or Gleason Score (R^2^ 0.02 and 0.29, respectively).

Bilen et al. [36] evaluated both 15 CRPC and 2 NEPC subjects with [^68^Ga]Ga-DOTATATE PET/CT in a prognostic setting. All patients demonstrated at least one avid lesion (mean SUVmax 12.19), and all seven patients with marked uptake were non-responders to systemic therapy and died in the follow-up, with a mean time to death of 8.1 months. In the group of six patients with moderate uptake, four died with a median time to death of 13.3 months, and of the surviving patients, none of them had the presence of NEPC. In the remaining group of three patients with mild uptake, all patients were still alive after 36 months of follow-up. Interestingly, the two patients with NEPC had higher SUVmax in comparison with the 14 subjects with non-NEPC CRPC (*p* value 0.04).

#### 3.2.3. PSMA

Sixty patients with liver metastases were evaluated with [^68^Ga]Ga-PSMA-11 PET/CT, in comparison with CT or MR, by Mattoni et al. [35] in a multicentric study and in 2/9 subjects that performed biopsy, the presence of NEPC was demonstrated. The overall detection rate of PET/CT was 92%, with a moderate and positive correlation between PSA and the presence of metastatic disease. Overall, the sensitivity, specificity, positive predictive value, negative predictive value, and accuracy were 58%, 92%, 82%, 77%, and 78%, respectively. The mean SUVmax of all PSMA-positive lesions was 20, significantly higher than the mean uptake of normal liver (*p* value < 0.01). A higher number of liver lesions was demonstrated in subjects with PSMA-positive PET compared to PSMA-negative scan (*p* value 0.013), and, in multivariate analysis, PSA and PET/CT were associated with the presence of liver metastases (*p* value < 0.01). Interestingly, the authors also build a radiomics model combining both CT and PET features for the detection of liver lesions, with an area under the curve of 0.807.

#### 3.2.4. Other Tracers and Study with Mixed Radiopharmaceuticals

Iravani et al. [33] performed an interesting study by comparing the results of [^68^Ga]Ga-PSMA-11, [^68^Ga]Ga-DOTATATE, and [^18^F]FDG PET/CT scans in 5 NEPC subjects, revealing that the median whole-body tumor volume was significantly higher for [^18^F]FDG (280 mL) compared to other radiopharmaceuticals (7 mL for [^68^Ga]Ga-PSMA-11 with a *p* value of 0.01 and 0 for [^68^Ga]Ga-DOTATATE with a *p* value of 0.02). Moreover, the median SUVmax values of the most avid lesion for [^68^Ga]Ga-PSMA-11, [^68^Ga]Ga-DOTATATE, and [^18^F]FDG were 13, 9.8 and 18.5, respectively. Three patients demonstrated both [^18^F]FDG and [^68^Ga]Ga-PSMA avidity, and for 2 of them, avidity of [^68^Ga]Ga-DOTATATE was also reported, even if spatial discordance for the lesions between the tracers was present, with [^18^F]FDG avid ones that lacked either [^68^Ga]Ga-PSMA and [^68^Ga]Ga-DOTATATE uptake. Interestingly, immunohistochemistry findings were consistent with both [^68^Ga]Ga-PSMA and [^68^Ga]Ga-DOTATATE results.

More recently, Vargas Ahumada et al. [37] performed an analysis on NEPC induced from CRPC with [^18^F]-PSMA-1007, [^18^F]-AIF-NOTA-octreotide and [^18^F]FDG, revealing that these tracers were able to assess the presence of bone, visceral, nodal and prostate lesions. Interestingly, [^18^F]-PSMA-1007 had the greater uptake compared to other radiopharmaceuticals: average SUVmax for [^18^F]-PSMA-1007, [^18^F]-AIF-NOTA-octreotide and [^18^F]FDG were 6.75, 4.6 and 6.4, respectively. Nevertheless, [^18^F]FDG was identified as the best tracer for the overall identification of such lesions, in particular for visceral localization. Moreover, 85/273 lesions detected were concordant between [^18^F]-PSMA-1007 and [^18^F]FDG, and the summation of both these imaging modalities reached a visualization rate of 98.9% of all the lesions present at CT. [^18^F]-AIF-NOTA-octreotide showed only a low detection rate.

Lastly, Kesch et al. [34] performed an interesting and innovative study with [^68^Ga]Ga-FAPI-04 PET/CT in a theranostic setting with immunohistochemistry, performing two scans on CRPC subjects underlying the presence of multiple bones and nodes metastatic lesions. In this setting, the mean SUVmax for bone, nodal, and lung metastases were 12.19, 12.58, and 6.30, respectively.

### 3.3. CI Studies

As mentioned, the role of CI in NEPC is limited due to the fact that it can underline features that are nonspecific and reflect the aggressiveness of these neoplasms. As a consequence, only a few studies have been published in this setting [38,39,40].

First, Schwartz et al. [38] evaluated the role of CT in the assessment of small cell (12 patients) and anaplastic PCa (15 subjects) and correlated its findings with PSA levels. The authors reported that, in the small cell cancer group with prostatic or abdominal masses, the mean PSA levels were less than 10 ng/mL, compared to less than 20 ng/mL for the group with small cell cancer and bone metastases.

More recently, He et al. [39] proposed an evaluation of 2 NEPC patients, underlying the fact that the US revealed the presence of an enlarged prostate while CT confirmed such insights, the presence of enhancing tissue, and eventually multiple metastases. MR characteristics of NEPC, performed; however, only in one patient, were the presence of a higher signal in T2-weighted imaging (T2WI) and diffusion-weighted imaging (DWI) sequences, arterial phase enhancement, irregular form, and/or the presence of metastases.

Lastly, Feng et al. [40] described the MR appearance of uncommon prostatic malignancies, including three subjects with small cell carcinoma. In this setting, T2WI imaging revealed a large heterogeneous, mildly hyperintense prostatic mass with an incomplete capsule, while T1WI imaging underlined an isointense lesion with mild or moderate heterogeneous enhancement after contrast agent injection. Moreover, invasion of adjacent structures (bladder, seminal vesicles, and rectum) and lymphatic or distant metastases were reported.

## 4. Discussion

NEPC is an aggressive form of PCa characterized by rapidly progressive symptoms, and it is locally advanced or metastatic at the time of diagnosis with frequent visceral metastasis, resulting in an extremely poor prognosis [6,7,8,9,10,11].

On the basis of these characteristics, imaging is mandatory for its work-up and assessment in order to evaluate disease extension and the possible presence of metastases. As mentioned, CI plays an important role in the evaluation of PCa andMR, CT, and US are useful for staging and restaging of classic forms of this tumor, even if their role in NEPC could be limited and not specific [15,16]. In particular, the role of CI in NEPC has been mostly evaluated in a staging setting in order to improve its preoperative diagnosis. Moreover, CI cannot directly differentiate NEPC from other PCa tumors since it gives only indirect information about the presence of necrosis, local invasion, metastases, or rapid progression [15,17]. As a consequence, only a small number of studies regarding the role of CI in NEPC have been published and were therefore included in the review, most of them with a limited sample of patients analyzed [38,39,40]. In general, CT revealed the ability to assess both primary NEPC and its metastases, while the role of MR seemed to have a relative ability to characterize this neoplasm, in particular for T2W1 and DWI sequences. However, data are still limited to draw a final conclusion on the role and the specific characteristics of CI in this rare form of neoplasm, and therefore, more studied are needed in this setting.

In contrast, the role of nuclear medicine imaging (and in particular PET/CT) in NEPC has been more widely studied [28,29,30,31,32,33,34,35,36,37]. In this setting, our review was performed considering different radiopharmaceuticals in order to assess their diagnostic usefulness for NEPC, and different mechanisms of uptake have been described for each tracer. Starting from [^18^F]FDG, it has been reported that NEPC is characterized by an elevation of glycolytic activity, resulting in an augmented expression of hexokinase and some specific glucose transporter (GLUT) isoforms with high affinity to the tracer [41]. Conversely, a reduction in PSMA expression as a result of augmented lineage plasticity related to AR inhibition in NEPC has been reported, and furthermore, this expression inversely correlates with the presence of markers of neuroendocrine differentiation [41,42]. Speaking of radiolabeled somatostatin analogs, it is known that normal prostatic tissue is characterized by physiological tracer uptake, but it has been demonstrated that neuroendocrine transdifferentiation can lead to both a reduction in PSMA levels and an augmented expression of somatostatin receptor (SSTR), in particular, SSTR2, that are a specific site of binding of such tracers [25,42]. Lastly, in the case of FAPI PET/CT, it has been demonstrated that FAPI expression increases with the progression of Pca, in particular, in CRPC that, as mentioned, can be associated with the development of NEPC [34].

[^18^F]FDG is the most common PET radiopharmaceutical used worldwide, and it is known that while tracer avidity is low in naive PCa, its uptake can be increased in CRPC and NEPC [3]. In this setting, three articles included in the review used [^18^F]FDG, and two of them compared it with other PET radiopharmaceuticals [29,33,37]. In general, it proved its ability to detect NEPC and its metastases, demonstrating a higher number of lesions compared to other tracers. Interestingly, some prognostic insights for [^18^F]FDG PET/CT were proposed. Nevertheless, it is important to underline that the mentioned findings on the role of [^18^F]FDG are based on the analyses of the accumulated clinical data in PCa patients rather than the analyses of the mechanism of NEPC. In this setting, this tracer cannot differentiate neuroendocrine differentiation from complex PCa cases by only relying on tumor uptake mechanism, and cases of positive PET/CT scans have been mainly described in treatment-induced NEPC [43]. Generally speaking, it has been reported that neuroendocrine differentiation is not always associated with [^18^F]FDG uptake, and moreover, PET/CT with this radiopharmaceutical is usually performed for high-grade and poorly differentiated forms of neuroendocrine neoplasms [3,44,45].

Another important class of positron emitters radiotracers particularly studied in NEPC are somatostatin analogs, given their ability to assess the presence of somatostatin receptors, possibly enabling the use of PRRT in these patients [25,26]. In this setting, [^68^Ga]Ga-DOTANOC has demonstrated good accuracy for the assessment of NEPC and CRPC and their metastases, even if sometimes scant uptake of such lesions has been reported [28,30,31]. Some studies also evaluated the role of [^68^Ga]Ga-DOTATATE PET/CT, revealing that this imaging modality was able to underline the presence of NEPC metastases. Interestingly, some insights on its prognostic value were also proposed in the setting of therapy response and survival. The ability to reveal the presence of NEPC was also demonstrated for PSMA PET/CT even if, compared to [^18^F]FDG, discordant findings have been described [33,35,37]. Interestingly, PSMA PET/CT demonstrated high specificity for the assessment of liver metastases in CRPC [35]. Lastly, a single study reported some insights on the usefulness of [^68^Ga]Ga-FAPI-04 PET/CT for the evaluation of bone and node metastases in NEPC [34].

Speaking about tracer uptake, based on the data reported in this review, it is hard to define clear evidence on which radiopharmaceutical has the greatest one. In particular, the studies included are characterized by a small cohort of patients, and in most of them, only the mean SUVmax values of all the patients for each tracer were reported. In general, a total of 36 subjects were evaluated with [^18^F]FDG, 53 with radiolabeled somatostatin analogs, 73 with PSMA, and 2 with [^68^Ga]Ga-FAPI-04. Overall labeled PSMA had the highest reported mean uptake (SUVmax 20) even if the range of mean SUVmax were really heterogeneous (18.5–4.52 for [^18^F]FDG, 12.19–1.57 for labeled somatostatin analogs, 20–6.75 for labeled PSMA and 12.58–5.90 for [^68^Ga]Ga-FAPI-04) to draw a final conclusion on tracer uptake.

As mentioned, some insights on the prognostic role of PET/CT in NEPC were proposed. Such findings were reported by Spratt et al. [29], who evaluated 23 NEPC with [^18^F]FDG, and by Bilen et al. [36], who imaged 17 patients with both metastatic CRPC or NEPC with [^68^Ga]Ga-DOTATATE. In the first case, patients with tracer-avid lesions or higher SUVmax had lower survival, while, in the second paper, subjects with marked uptake were non-responders to systemic therapy and died in the follow-up. Based on these findings, on the number of subjects included in both studies, and on the fact that the second paper also focused on CRPC subjects, clear evidence on the prognostic role of PET/CT in NEPC is not available, and further research is needed in this field.

Even if not included in this review, data published in the literature have also underlined the role of PET/CT imaging in PCa and NEPC in preclinical settings. In this scenario, it has been proposed that specific forms of NEPC, depending on their origin, can be imaged with different radiopharmaceuticals based on preclinical evidence [41,43,46,47]. Moreover, new tracers able to evaluate the presence of NEPC or its progression from CRPC are continuously developed and hopefully will help in the assessment of the disease [34,48,49,50]. Lastly, some interesting insights on the role of these new radiopharmaceuticals for the evaluation of treatment response with innovative drugs specific to NEPC have been reported [51].

As previously underlined, NEPC can develop as a de novo form arising from normal prostatic neuroendocrine cells or can derive after neuroendocrine transdifferentiation of PCa in the case of CRPC. These two tumoral entities are characterized using different clinical behavior and prognosis, and therefore, an imaging diagnostic tool able to characterize their differences would be helpful in the clinical management of such patients. However, our review is not able to clearly identify discrepancies in terms of PET/CT performances with different radiopharmaceuticals between de novo and derived NEPC since only a small amount of patients with the first type were included in the studies selected for the review and, therefore, clear evidence in this setting can not be underlined.

This review is not without limitations, and one of the most important is the fact that most of the studies included were performed only with small samples of patients, a fact that needs to be correlated with the rarity of NEPC. Furthermore, some of these studies were performed with heterogeneous cohorts, with the inclusion not only of patients with NEPC but also of CRPC subjects. Another significant limitation is that we evaluated the analyses of the accumulated clinical data in prostate cancer patients rather than the analyses of the mechanism of NEPC. In this setting, the insights reported by this review need to be confirmed using wider and multicentric studies.

## 5. Conclusions

In conclusion, published data on the role of PET/CT and CI for the assessment of NEPC are limited. In this setting, at present, it is still uncertain which tracer performs best, and although [^18^F]FDG has been evaluated and seems to offer some advantages in availability and clinical staging, other tracers may be more useful to understand tumor biology or identifying a target for subsequent radioligand therapy. Moreover, it should be underlined that most of the available data concerned labeled somatostatin analogs, while only two papers evaluated the role of [^18^F]FDG in this rare neuroendocrine neoplasia. Further research is therefore desirable, and the possibility to compare different tracers in future studies should be considered. In contrast, data are still limited to draw a final conclusion on the role and the specific characteristics of CI in this rare form of neoplasm, and therefore, more studies are needed in this setting.

## Figures and Tables

**Figure 1 cancers-15-04404-f001:**
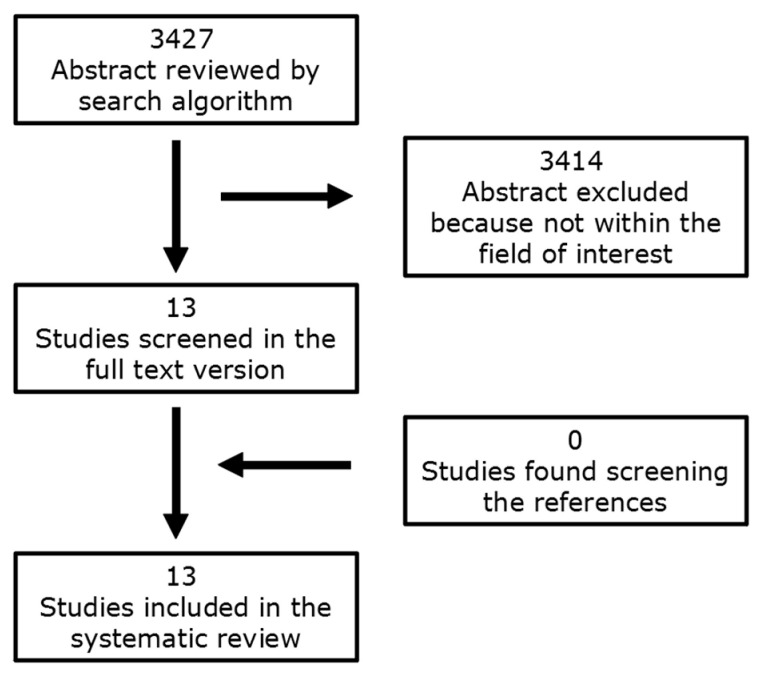
Flowchart of the research flow of eligible studies on the role of PET/CT and CI for the assessment of neuroendocrine prostate cancer.

**Figure 2 cancers-15-04404-f002:**
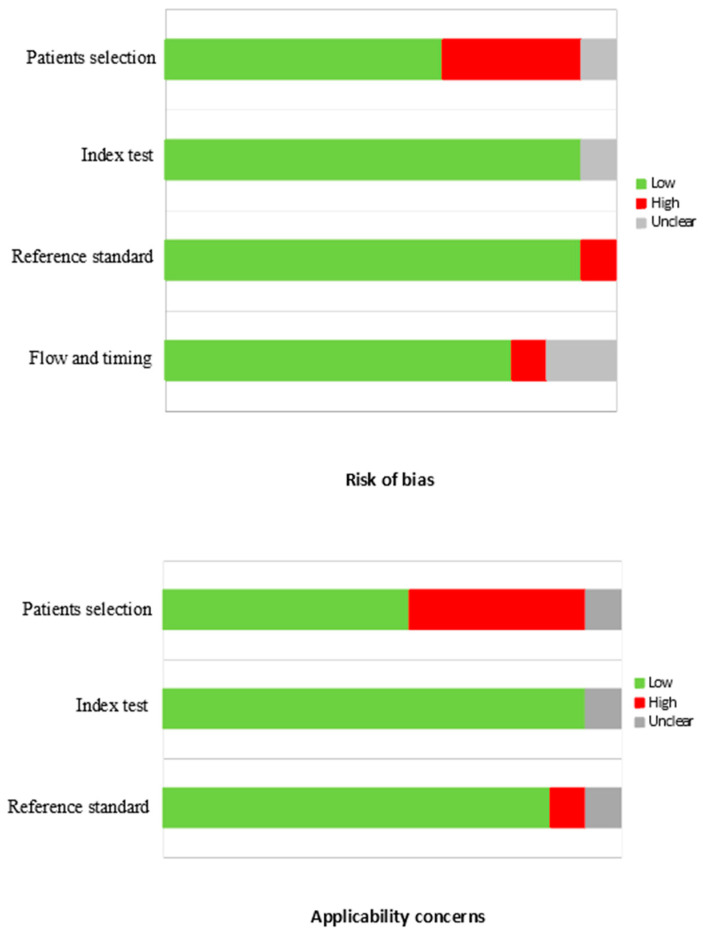
QUADAS-2 quality assessment for risk of bias and applicability concerns for the studies considered in the review.

**Table 1 cancers-15-04404-t001:** Characteristics of the studies selected and considered for the systematic review.

First Author	Ref. N.	Year	Country	Study Design	Tracers	Type of Scan	N. Pts. Scanned	Average SUV	Setting	Main Findings
**PET imaging**										
Fanti S.	[28]	2008	Italy	Retrospective	[^68^Ga]Ga-DOTANOC	PET/CT	3	ns	Suspected relapse, therapy planning, or previous indeterminate findings of NET	[^68^Ga]Ga-DOTA-NOC can be usefully applied for the evaluation of NET of uncommon presentation
Spratt D.E.	[29]	2014	USA	Retrospective	[^18^F]FDG	PET/CT	23	4.52 for bone lesions, 6.65 for soft tissue lesions	Assessment of CRPC with high serum tumor markers	[^18^F]FDG has clinical utility in the detection and monitoring of soft tissue and bone metastases
Savelli G.	[30]	2014	Italy	Prospective	[^68^Ga]Ga-DOTANOC	PET/CT	2	1.57	Assessment of somatostatin overexpression in NEPC	[^68^Ga]Ga-DOTANOC PET/CT can visualize NEPC localization
Savelli G.	[31]	2015	Italy	Prospective	[^68^Ga]Ga-DOTANOC	PET/CT	6	ns	Assessment of CRPC	[^68^Ga]Ga-DOTANOC has the potential to evaluate some CRPC
Gofrit O.N.	[32]	2017	Israel	ns	[^68^Ga]Ga-DOTATATE	PET/CT	12	5.3 for blastic lesion, 7.2 for lytic or nodal metastases	Assessment of CRPC	[^68^Ga]Ga-DOTATATE uptake is a common finding in CRPC metastases
Iravani A.	[33]	2021	Australia	Retrospective	[^68^Ga]Ga-PSMA, [^68^Ga]Ga-DOTATATE, [^18^F]FDG	PET/CT	5	13.0 for [^68^Ga]Ga-PSMA, 9.8 for [^68^Ga]Ga-DOTATATE, 18.5 for [^18^F]FDG	Pre-radioling therapy evaluation	NEPC has wide inter- and intrapatient molecular imaging heterogeneity by the different tracers
Kesch C.	[34]	2021	Germany, Hungary, Canada	Retrospective	[^68^Ga]Ga-FAPI-04	PET/CT	2	7.17 for bone metastases	Assessment of CRPC	Increased FAP tissue expression in CRPC
Mattoni S.	[35]	2022	Italy, Argentina, Brazil, Germany	Retrospective	[^68^Ga]Ga-PSMA-11	PET/CT, PET/MR	60	20	Assessment of liver metastasis in CRPC	[^68^Ga]Ga-PSMA-11 has high specificity, positive predictive value, and reproducibility compared to CI and liver biopsy when assessing unfavorable liver metastases in CRPC patients
Bilen M.A.	[36]	2022	USA	Retrospective	[^68^Ga]Ga-DOTATATE	PET/CT	17	12.19	Assessment of CRPC	[^68^Ga]Ga-DOTATATE is able to identify CRPC and NEPC metastatic deposits and lesions; high uptake may portend poor outcomes
Vargas Ahumada J.	[37]	2022	Colombia, Mexico	Retrospective	[^18^F]-PSMA-1007, [^18^F]-AIF-NOTA-octreotide, [^18^F]FDG	PET/CT	8	6.75 [^18^F]-PSMA-1007, 4.60 [^18^F]-AIF-NOTA-octreotide, 6.40 [^18^F]FDG	Assessment of CRPC	NEPC has wide inter- and intrapatient heterogeneity. [^18^F]FDG detected most lesions, even though [^18^F]-PSMA-1007 detected more bone lesions
**Conventional imaging**										
Schwatrz L.H.	[38]	1998	USA	Retrospective	/	CT	27 *	/	Pre-chemotherapy assessment	Patients with the anaplastic clinical variant of prostate cancer often have extensive metastatic disease at CT despite relatively low PSA levels
He H.Q.	[39]	2015	China	Retrospective	/	US, CT, MR	2	/	Diagnosis of NEPC	MR, CT, self-contradictory clinical appearance, and laboratory results can help to achieve an accurate diagnosis of PNEC
Feng Z.Y.	[40]	2017	China	Retrospective	/	MR	3	/	Staging	Some differences among PNEC and other uncommon prostatic tumors are present, although some overlap in the MR imaging were reported

* 15 anaplastic prostate cancer and 12 small cell cancer of the prostate. N.: number; NEPC: neuroendocrine prostate cancer; Pts: patients; Ref: reference; NET: neuroendocrine tumor; NED: neuroendocrine differentiation; CRCP: castration-resistant prostate cancer; PET/CT: positron emission tomography/computed tomography; PSA: prostate-specific antigen; MR: magnetic resonance; US: ultrasound, CI: conventional imaging; [^18^F]FDG: [^18^F]fluorodeoxyglucose; PSMA: prostate-specific membrane antigen; SUV: standardized uptake value; ns: not specified.

## Data Availability

Data supporting the reported results can be found using the public PubMed/MEDLINE, Scopus and Web of Science databases.

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
