# Peer review of "PET/CT and Conventional Imaging for the Assessment of Neuroendocrine Prostate Cancer: A Systematic Review"

_cancers, 2023, doi:10.3390/cancers15174404_

Round 1

Reviewer 1 Report (Previous Reviewer 2)

Line 23: Limited published data in this setting are present can be deleted,

Line 37 : Should read - PET/CT imaging with different radiopharmaceuticals has been evaluated in a many studies  not "higher" number of studies

The abstract seems incomplete. Please provide clean document without track changes

Line 61: Can arise denovo - not as a de novo form

The review is still long. For example line 428 - 436 are superfluous

English needs to be improved

Author Response

Dear Reviewers,

thank you for the evaluation of the manuscript and the useful comments that you underlined and that we carefully considered.

In particular, for Reviewer 1:

- “Line 23: Limited published data in this setting are present can be deleted”: we deleted this sentence according to this suggestion;

- “Line 37 : Should read - PET/CT imaging with different radiopharmaceuticals has been evaluated in a many studies not "higher" number of studies”: the text was corrected according to this suggestion;

- “The abstract seems incomplete. Please provide clean document without track changes”: the results section of the abstract was expanded with more data and we also uploaded a version of the manuscript without track changes in order to have a clear view of the whole abstract;

- “Line 61: Can arise denovo - not as a de novo form”: the text was corrected according to this suggestion;

- “The review is still long. For example line 428 - 436 are superfluous”: we agree with the reviewer on the fact that the review is long, however since this is the fourth revision of the paper many parts were added based on the suggestions of different reviewers. In particular, the lines 428-436 were added as a response to a clear request by a previous reviewer.

- English language was improved through the text.

Hoping that our changes could lead to the acceptance of the manuscripts,

thank you again and best regards.

The authors

Reviewer 2 Report (New Reviewer)

The paper is well organized and cites QUADS 2 for quality assessment for risk of bias and applicability issues for the studies considered in the review. It also provides an example of the application of the tool. The addition of the SUV and Tracer columns has been essential. In the conclusions, it could be underlined that most of the available data concerned DOTA tracers, and only in 2 papers is FDG used to characterize this rare neuroendocrine neoplasia. Another suggestion involves the possibility of using both tracers in future studies.

The English used is appropriate to the subject matter.

Author Response

Dear Reviewers,

thank you for the evaluation of the manuscript and the useful comments that you underlined and that we carefully considered.

In particular, for Reviewer 2:

- “In the conclusions, it could be underlined that most of the available data concerned DOTA tracers, and only in 2 papers is FDG used to characterize this rare neuroendocrine neoplasia”: the text was corrected according to this suggestion;

- “Another suggestion involves the possibility of using both tracers in future studies.” the text was corrected according to this suggestion and this part was added in the conclusion.

Hoping that our changes could lead to the acceptance of the manuscripts,

thank you again and best regards.

The authors

Reviewer 3 Report (New Reviewer)

The authors have diligently addressed the comments and suggestions provided by both the reviewers and editors, resulting in significant improvements to the manuscript. Their revisions demonstrate a high level of responsiveness and commitment to enhancing the quality of their work. Based on the thorough revisions made, I am of the opinion that the manuscript is now suitable for publication in its current form.

none

Author Response

Dear Reviewers,

thank you for the evaluation of the manuscript and the useful comments that you underlined and that we carefully considered.

In particular, for Reviewer 3:

Dear Reviewer, thank you again for the evaluation of this version of the manuscript and for its approval.

Hoping that our changes could lead to the acceptance of the manuscripts,

thank you again and best regards.

The authors

Round 2

Reviewer 1 Report (Previous Reviewer 2)

The document is suitable for publication

This manuscript is a resubmission of an earlier submission. The following is a list of the peer review reports and author responses from that submission.

Round 1

Reviewer 1 Report

This review manuscript focuses on discussing current PET/CT and COI for assessing NEPC. The chosen topic is interesting to the prostate cancer research field. The authors pointed out the limited progress in the PET imaging of NEPC and stated that the most promising tracer seems to be FDG. The conclusion was made based on the analyses of the accumulated clinical data in prostate cancer patients rather than the analyses of the mechanism of NEPC, which is one of the significant limitations. FDG cannot differentiate neuroendocrine differentiation from complex prostate cancer cases by looking at its tumor uptake mechanism. The other limitation is that the authors should have selected and summarized publications with preclinical studies in NEPC, because it excludes some latest and exciting findings in this specific topic.

Other comments are as below:

1.     Page 3, line 123: please correct “Pca”.

2.     Page 3, line 136: please use [177Lu]Lu-PSMA-617.

3.     Page 3, line 136: please expand “different conditions” to make the statement clear.

4.     Page 8, line 210: please correct “CRCP”.

5.     Page 8, line 213: please use superscript format for 18 in 18F.

6.     Page 9, line 270: please use [68Ga]Ga-PSMA-11.

7.     Page 10, line 328: please correct “relatively”. 

The quality of the English Language is overall good. 

Author Response

Dear reviewer,

thank you for your precise and useful comments, we carefully considered all of them and made the changes that you requested. In particular:

- “The conclusion was made based on the analyses of the accumulated clinical data in prostate cancer patients rather than the analyses of the mechanism of NEPC, which is one of the significant limitations.”: we included a brief discussion of this limitation in the discussion section and we also add it to the limitations of the study (lines 343-351 and lines 379-381).

- “the authors should have selected and summarized publications with preclinical studies in NEPC, because it excludes some latest and exciting findings in this specific topic.”: we strongly agree with you that preclinical studies are fundamental in order to understand more about NEPC. However, we stated in the materials and methods section that preclinical studies were excluded from the review, therefore we can not add them after performing the analyses. In order to avoid this limitation, we discussed the role of PET/CT in preclinical setting in the discussion section (lines 366-374).

- Other comments about writing mistakes were corrected according to your suggestion.

Best regards,

the authors.

Reviewer 2 Report

The article is interesting. It is trifle too long with repetition of concepts eg in line 72 -75 is being repeated again in lines 98. -102. The english also requires extensive review and is gramatically wrong at multiple places

The english also requires extensive review and is gramatically wrong at multiple places

Author Response

Dear reviewer,

thank you for your precise and useful comments, we carefully considered all of them and made the changes that you requested. In particular:

- the introduction of the paper was revised and shortened as you suggested.

- the manuscript was completely revised in order to improve the quality of the english language.

Best regards,

the authors.

Reviewer 3 Report

This is an excellent systematic review of the role of PET in neuroendocrine prostate cancer. The methods are thoroughly described, and the studies were discussed at an excellent level of detail. No major scientific inaccuracies were detected. As the role of PET within prostate cancer management continues to increase, the selection of this topic for a systemic review is very timely.

English language is excellent throughout. Minor editing required in the first sentence of the Simple Summary mentions "neuroendocrine prostate of the cancer," which should be "neuroendocrine prostate cancer."

Author Response

Dear reviewer,

thank you for your review and compliments.

Best regards,

the authors.

Round 2

Reviewer 1 Report

Thanks for the response and additions to the manuscript. However, the paper still focuses on discussing the role of FDG in NEPC, which does not improve the overall value.